# Emergent Graphical Conventions in a Visual Communication Game

**Shuwen Qiu**[*,1], **Sirui Xie**[*,1], **Lifeng Fan**[2],
**Tao Gao**[3,4], **Jungseock Joo**[3], **Song-Chun Zhu**[1,2,4,5], **Yixin Zhu**[5]

[1] Department of Computer Science, UCLA
[2] Beijing Institute for General Artificial Intelligence (BIGAI)
[3] Department of Communication, UCLA    [4] Department of Statistics, UCLA
[5] Institute for Artificial Intelligence, Peking University

https://sites.google.com/view/emergent-graphical-conventions

## Abstract

Humans communicate with graphical sketches apart from symbolic languages (Fay et al., 2014). Primarily focusing on the latter, recent studies of emergent communication (Lazaridou and Baroni, 2020) overlook the sketches; they do not account for the evolution process through which symbolic sign systems emerge in the trade-off between iconicity and symbolicity. In this work, we take the very first step to model and simulate this process via two neural agents playing a visual communication game; the sender communicates with the receiver by sketching on a canvas. We devise a novel reinforcement learning method such that agents are evolved jointly towards successful communication and abstract graphical conventions. To inspect the emerged conventions, we define three fundamental properties—iconicity, symbolicity, and semanticity—and design evaluation methods accordingly. Our experimental results under different controls are consistent with the observation in studies of human graphical conventions (Hawkins et al., 2019; Fay et al., 2010). Of note, we find that evolved sketches can preserve the continuum of semantics (Mikolov et al., 2013) under proper environmental pressures. More interestingly, co-evolved agents can switch between conventionalized and iconic communication based on their familiarity with referents. We hope the present research can pave the path for studying emergent communication with the modality of sketches.

## 1 Introduction

Communication problem naturally arises when traveling in a foreign country where you do not speak the native language, which necessitates exploring non-linguistic means of communication, such as drawings. Due to its *iconic* nature (*i.e.*, perceptual resemblance to or natural association with the referent), drawings serve as a powerful tool to communicate concepts transcending language barriers (Fay et al., 2014). In fact, we humans started to use drawings to convey messages dating back to 40,000–60,000 years ago (Hoffmann et al., 2018; Hawkins et al., 2019). Some cognitive science studies hypothesize a transition from sketch-based communication before the formation of sign systems and provide evidence that iconic signs can gradually become *symbolic* through repeated communication (Fay et al., 2014). In contrast to *icons*, *symbols* are special forms bearing arbitrary relations to the referents. Fig. 1 describes a typical scenario of such phenomena: Alice (in green) uses a sketch to communicate the concept "rooster" to Bob (in yellow). Initially, they need to ground the sketch to the referent. Later, details of the visual concept, such as strokes of the head and body, are gradually abstracted away, leaving only the most salient part, the crown. The iconicity in the communicated sketch drops while the symbolicity rises.

---

[*] indicates equal contribution.

36th Conference on Neural Information Processing Systems (NeurIPS 2022).

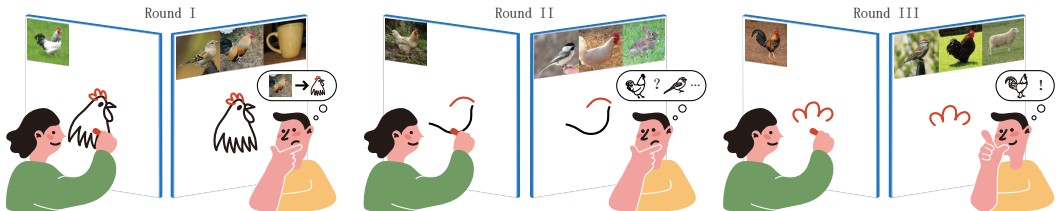

Figure 1: **An example of the visual communication game.** In an iterative sketch communication game (Hawkins et al., 2019), players first need to ground sketches to referents. The drawer (Alice) gradually simplifies the drawing but keeps the most salient parts of the target concept (rooster crown). This evolution process enables the viewer (Bob) to promptly distinguish the target (rooster) from distractors (bird, cup, rabbit, and sheep).

While models of emerging communication protocols has attracted attention (Lazaridou et al., 2017; Cao et al., 2018; Evtimova et al., 2018; Havrylov and Titov, 2017; Lazaridou et al., 2018; Lazaridou and Baroni, 2020; Mordatch and Abbeel, 2018; Ren et al., 2020; Eccles et al., 2019; Graesser et al., 2019), the initial and primary communication medium is presumed and limited to be symbolic rather than iconic. By simulating a multi-agent *referential game*, prior work seeks for the *environmental driving forces* behind the emergence of effective communications. In a typical setup of referential games, two agents play similar roles as in the above Alice-Bob example but share a primitive set of arbitrary tokens (*i.e.*, the vocabulary). Using these tokens, an agent (the sender) attempts to communicate a message to another agent (the receiver). A communication convention emerges when two agents successfully communicate by associating visual concepts in the images with tokens in the pre-selected vocabulary. Though this line of work has probed into some linguistic properties of the established communication conventions (Lazaridou et al., 2017; 2018; Ren et al., 2020), some intriguing questions remain open: How do agents make a trade-off between iconicity and symbolicity to emerge symbolic sign systems?

In this work, we present the very first step of modeling and simulating the evolution process of *graphical conventions* (Hawkins et al., 2019), a two-participant communication convention whose medium is drawings in an abstract form. Specifically, we consider our **contributions** in three folds:

First, we model a multi-agent visual communication game and propose a learning framework, wherein the sender and the receiver evolve jointly. This visual communication game is an alternating sequential decision-making process, in which the sender generates a sequence of strokes step by step, terminated by the receiver. In contrast to discretized tokens in prior work, strokes can be continuously parametrized (Ha and Eck, 2018; Huang et al., 2019) such that the derivatives of learning objectives can be more effectively back-propagated through communication channels (Foerster et al., 2016). We further derive a novel training surrogate for multi-agent reinforcement learning based on a joint value function and the eligibility trace method (Sutton and Barto, 2018). This differs from the REINFORCE-based surrogate in prior works on early-terminable sequential communication (Cao et al., 2018; Evtimova et al., 2018) and the TD learning method in the literature of multi-agent turn-taking games, wherein each agent has its own value function and has to model the value of others (Wen et al., 2019). In experiments, we empirically demonstrate that our integration of function approximation and Monte Carlo sampling facilitates the agents' awareness of the correlation between complex and simple sketches, thereby enabling a smooth abstraction process.

Second, we define essential properties in studying evolved sketches. Specifically, we define *iconicity* (Fay et al., 2014) as the drawings exhibiting high visual resemblance to the corresponding images, such that they are proximal to the latter when measured on the high-level embedding of a general-purpose visual system; we define *symbolicity* (Fay et al., 2018) as these drawings being consistently separable in the high-level visual embedding, which facilitates new communication participants to easily distinguish between categories without grounding them to referents; and we define *semanticity* (Harispe et al., 2015) as the topography of the high-level embedding space of the drawings being strongly correlated to that of images, such that semantically similar instances and categories lie close to each other in the embedding space. Of note, this is not the only way to define these cognitive concepts; we intend to align readers on critical concepts in our work.

Third, we present a suite of quantitative and qualitative methods to evaluate the emergent graphical conventions based on the above carefully defined *iconicity*, *symbolicity*, and *semanticity*. This is necessary because a high communication rate does not imply good representations (Bouchacourt and Baroni, 2018). The graphical nature of the communication medium mandates us to repurpose

representation learning metrics rather than adopt linguistic metrics in emergent symbolic communication. We evaluate the contribution of different environmental drivers, early decision, sender's update, and sequential communication, to the three properties of the emergent conventions. Critically, the empirical results assessed on our metrics align well with our prediction based on the findings of human graphical conventions (Fay et al., 2010; Hawkins et al., 2019), justifying our environment, model, and evaluation. One of these setups emerges conventions where the three properties are consistent with our expectation of a sign system. Particularly, we find two inspiring phenomena: (i) Evolved sketches from semantically similar classes are perceptually more similar to each other than those falling into different superclasses. (ii) To communicate concepts not included in their established conventions, evolved agents can return to more iconic communications than humans. We hope our work sheds light on emergent communication in the modality of sketches and facilitates the study of cognitive evolutionary theories of pictographic sign systems.

## 2 Related work

**Learning to sketch**  Ha and Eck (2018) begin the endeavor of teaching modern neural models to sketch stroke by stroke. However, generating meaningful stroke sequences directly from various categories of natural images is still in the early phase (Song et al., 2018; Wang et al., 2021; Zou et al., 2018). To assure the interestingness of the category-level sketch communication, we design a stage-wise agent that transfers a natural image into a pixel-level sketch (Kampelmuhler and Pinz, 2020) and draws the sketch on the canvas stroke by stroke with a policy (Huang et al., 2019). To pre-train our neural agents to sketch, we select Sketchy (Sangkloy et al., 2016) from many datasets (Yu et al., 2016; Ha and Eck, 2018; Eitz et al., 2012; Sangkloy et al., 2016) for its fine-grained photo-sketch correspondence, rich stroke-level annotations, and well-organized categorization structures.

**Communication games**  While learning to sketch is formulated as a single-agent task with explicit supervision, our focus is on **how sketches would evolve** when utilized as the communication medium between *two cooperative agents*. Their cooperation is always formulated as a communication game, recently adopted to study phenomena in human-robot teaming (Yuan et al., 2022) and natural languages, such as symbolic language acquisition (Graesser et al., 2019) and the emergence of compositionality (Ren et al., 2020). Some notable works (Lazaridou et al., 2017; 2018; Evtimova et al., 2018; Havrylov and Titov, 2017) have devised interesting metrics, such as *purity* (Lazaridou et al., 2017) and *topographic similarity* (Lazaridou et al., 2018). In comparison, our work is unique due to the distinctive communication medium, continuously parametrized sketches. Although a concurrent work (Mihai and Hare, 2021) also enables the agents to sketch in a communication game, it focuses primarily on drawing interpretable sketches without abstracting them into graphical symbols along the communication. We position our work as an alternative to emergent symbolic communication, since the emergent graphical symbols may better represent the continuum of semantics, as encoded in the vector representation of tokens (Mikolov et al., 2013). Methodologically, we devise new evaluation metrics for sketch modality, assessing *iconicity*, *symbolicity*, and *semanticity* in the evolved sketches.

**Emergent graphical conventions**  Evolving from sketches to graphical conventions/symbols is an active field in cognitive science under the banner of "emergent sign systems." Fay et al. (2010) show that pair interaction can form their local conventions when they play the Pictionary game. Using natural images instead of texts as the prompt, Hawkins et al. (2019) show that visual properties of the images also influence the formed graphical conventions besides partners' shared interaction history; *i.e.*, the evolved sketches highlight visually salient parts. Nevertheless, only a few computational models exist apart from these human behavior studies. Fan et al. (2020) describe a model for selecting complex or simple sketches considering the communication context. Bhunia et al. (2020) and Muhammad et al. (2018) consider stroke selection and reordering to simplify the sketch. In contrast to sketch or stroke selection, we model **embodied** agents who can draw and recognize sketches.

## 3 The visual communication game

Our visual communication game is formulated as a tuple
$$(\mathcal{I}, C, \mathcal{A}_S, \mathcal{A}_R, G, r, \gamma),$$
where $\mathcal{I}$ is the image set to be presented to the sender $S$ and the receiver $R$. These images contain a single object in the foreground, and hence the image space $\mathcal{I}$ can be partitioned into $N$ classes

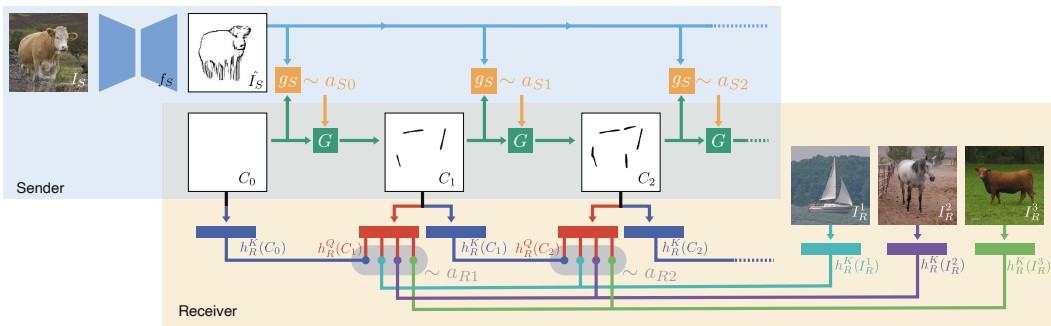

Figure 2: **Communication process**. In our visual communication game, a sender $S$ and a receiver $R$ only share the observation of the canvas $C$. The sender converts the natural image $I_S$ to a pixel-level sketch $\hat{I}_S$. At each step, the sender first draws five strokes $a_S$ through the renderer $G$, which updates the canvas to $C_{t+1}$. Next, the receiver uses the updated canvas $C_{t+1}$ to query from the context images $\{I_R^1, ..., I_R^M\}$ and the last canvas $C_t$, deciding the action $a_R$ at this step. The game continues if the receiver chooses to wait.

according to the object categories. In each game round, the sender is presented with one image $I_S$, and the receiver is presented with $M$ images $\{I_R^1, ..., I_R^M\}$. Prior work shows that category-level context can force the agent to coordinate on more abstract properties and help draw sketches to be more recognizable as the type of object (Lazaridou et al., 2017; Mihai and Hare, 2021). We make the observations of $S$ and $R$ disjoint (*i.e.*, $I_S \notin \{I_R^1, ..., I_R^M\}$) but with a target image $I_R^*$ in the same class as $I_S$. We refer to the $M$ images that the receiver can see as the *context*. Neither the receiver or the sender would see the image(s) presented to their partner; they can only communicate this information by drawing sketches on the canvas $C$, observable to both players. As shown in Fig. 2, $C_0$ is initialized to be blank at the beginning of each round. Only the sender can draw on the canvas with actions chosen from $\mathcal{A}_S$. The action at each time step consists of 5 strokes, which are continuous vectors in $\mathbb{R}^6$. We constrain each dimension to be in $(0, 1)$ due to the limited space of the canvas. The canvas is updated from $C_t$ to $C_{t+1}$ by the renderer $G$ after each step of the sender's sketching. The receiver, after observing the updated canvas, would choose among the $M$ images or wait for the next step from the sender; these $M + 1$ possible actions constitute $\mathcal{A}_R$. A game round terminates when the receiver gives up waiting and chooses one from the images. After the termination, the sender and the receiver will receive a shared reward or penalty, depending on if the receiver makes the right choice:

$$r : \mathcal{I} \times \mathcal{I} \to \{-1, 1\}.$$

This reward/penalty is temporally decayed with a decay factor $\gamma$. That is, if the receiver decides to choose from the images at step $t$, this cooperating pair will receive either $\gamma^t$ or $-\gamma^t$. Hence, even though the players do not receive an explicit penalty for long conversations, there is an implicit penalty/reward for delayed positive/negative answers. No reward will be assigned if the receiver chooses to wait. The next round starts after the reward/penalty assignment.

## 4 Agents

The two agents involved in the visual communication game are modeled with two decision policies, $\pi_S$ and $\pi_R$, for the sender and the receiver, respectively. These policies are stochastic mapping from the agents' observation space to the action space:

$$\pi_S : \mathcal{I} \times C \to \mathcal{P}(\mathcal{A}_S), \quad \pi_R : \mathcal{I}^M \times C \to \mathcal{P}(\mathcal{A}_R), \tag{1}$$

where $\mathcal{P}(\mathcal{A})$ is a distribution over the support set $\mathcal{A}$. As shown in Fig. 2, at each time step $t \in \{0, ...T\}$, the sender first emits the stroke parameters for the next five strokes $a_{St} \sim \pi_S(I_S, C_t)$. These strokes are applied to the canvas by a differentiable renderer, $C_{t+1} = G(C_t, a_{St})$. Next, the updated canvas is transmitted to the receiver. The receiver decides whether to terminate the game by making its prediction (*i.e.*, $a_{Rt} \in \{1, ..., M\}$) or wait for another round (*i.e.*, $a_{Rt} = M + 1$); its decision is sampled from $\pi_R$. If a prediction is made, it is used to select the image $I_R^{a_R}$ from $\{I_R^1, ...I_R^M\}$ and score this game with $r(I_S, I_R^{a_R})$. Otherwise, this routine repeats in the next step $t \leftarrow t + 1$.

## 4.1 Sender

Prior to playing the visual communication game, the sender should be able to (i) extract edges from natural images (Xie and Tu, 2015) and (ii) draw sketches that closely resemble the configurations of salient edges (Li et al., 2019), just as humans do (Sayim and Cavanagh, 2011). To endow the sender with these capabilities, we design a stage-wise architecture $h_S = g_S \circ f_S$. Specifically, $I_S$ is first converted to a target sketch $\hat{I}_S$ using a visual module $f_S$ (Kampelmuhler and Pinz, 2020), capturing the salient edge information in the natural image; we directly adopt the pre-trained model from the referred work. Next, $\hat{I}_S$ is concatenated with the current canvas $C_t$ and fed to the sketching module $g_S$, whose architecture is built upon Huang et al. (2019). This sketching module outputs five vectors in the form $(x_0, y_0, x_1, y_1, x_2, y_2)$, which parametrizes the curve of one stroke. The policy is parametrized as a Gaussian distribution during training,

$$\pi_S = \mathcal{N}(\mu_t, \sigma^2), \quad \mu_t = h_S(I_S, C_t), \quad \sigma^2 = c \cdot \mathbf{I}, \tag{2}$$

where $\mathbf{I}$ is the identity matrix, and $c$ is a constant hyperparameter. During the testing, we set $c = 0$.

These stroke parameters $a_{St}$ are fed into a pre-trained renderer $G$ (Huang et al., 2019) to update the canvas, $C_{t+1} = G(C_t, a_{St})$. This renderer is fully differentiable, enabling end-to-end model-based training (Hafner et al., 2019) of the sketching module $g_S$. We pre-train $g_S$ on Sketchy (Sangkloy et al., 2016); please refer to the supplementary video for results.

## 4.2 Receiver

The receiver, similar to the sender, should also carry some rudimentary visual capability to this game. Unlike the low-level vision needed for the sender, the requirement for the receiver is high-level visual recognition. Therefore, we adopt a pre-trained VGG16 (Simonyan and Zisserman, 2015) as the visual module $f_R : \mathcal{I} \rightarrow \mathbb{R}^{4096}$ of the receiver, following a similar practice in recent literature (Lazaridou et al., 2017; Havrylov and Titov, 2017). The output of this visual module is a vector, and further transformed by two separate linear layers, $g_R^K$ and $g_R^Q$, into visual embeddings, $h_R^K(I)$ and $h_R^Q(I)$. That is, $h_R^K = g_R^K \circ f_R, h_R^Q = g_R^Q \circ f_R$.

When observing both the context $\{I_R^1, ..., I_R^M\}$ and the canvas $C_t$, the receiver first embeds each of them with $h_R$. Next, it makes the decision based on the similarity between the current canvas and each option in the context. The decision module is thus realized by a Boltzmann distribution based on resemblance:

$$\pi_R(a_{Rt}|I_R^1, ... I_R^M, C_{t-1}, C_t) = \frac{\exp(h_R^Q(C_t) \cdot h_R^K(I_R^{a_{Rt}}))}{\displaystyle\sum_{m=1}^{M+1} \exp(h_R^Q(C_t) \cdot h_R^K(I_R^m))}, \tag{3}$$

where $I_R^{M+1} = C_{t-1}$. Of note, although a similar policy was proposed before (Lazaridou et al., 2018; Havrylov and Titov, 2017), our $\pi_R$ is distinct as it is endowed with an external memory of $C_{t-1}$. Intuitively, if the receiver finds the current canvas $C_t$ closer to the last canvas $C_{t-1}$ in the embedding space than all $M$ options in the context, it will choose to emit $a_{Rt} = M + 1$ and delay the decision to the next step; a prediction can only be made when the receiver finds the current canvas is informative enough. As a result, the sender would draw informative strokes as early as possible to avoid the implicit penalty in the decayed reward.

## 4.3 Learning to communicate

Policies of the sender and the receiver are trained jointly to maximize the objective:

$$\pi_S^*, \pi_R^* = \arg\max_{\pi_S, \pi_R} \mathbb{E}_{\tau \sim (\pi_S, \pi_R)} \left[ \sum_t \gamma^t r_t \right], \tag{4}$$

where $\tau = \{C_0, a_{S0}, C_1, a_{R1}, a_{S1}, ...\}$ is the simulated episodic trajectory. As well known in reinforcement learning, the analytical expectation in Eq. (4) is intractable to calculate along the trajectory $\tau$. We devise value functions $\mathcal{V}(X_t)$ and $V_\lambda(X_t)$ for an optimization surrogate:

$$\mathcal{V}(X_t) = \mathbb{E}_{\pi_S(a_{St}|I_S, C_{t-1}), \pi_R(a_{Rt}|\hat{X}_t)} \left[ (r_t + \gamma\delta(a_{Rt})V_\lambda(X_{t+1}) \right], \tag{5}$$

where $\delta(a_{Rt})$ is the Dirac delta function that returns 1 when the action is *wait* and 0 otherwise. $X_t = \text{cat}([I_S], \hat{X}_t)$, where $\hat{X}_t = [I_R^1, ... I_R^M, C_{t-1}, C_t]$. The expectation $\mathbb{E}_{\pi_S(a_{St}|I_S, C_{t-1})}[\cdot]$ is approximated with the point estimate, as in the reparametrization in VAE (Kingma and Welling, 2014). The expectation $\mathbb{E}_{\pi_R(a_{Rt}|\hat{X}_t)}[\cdot]$ can be calculated analytically because $\pi_{Rt}$ is a categorical distribution. The connection between $\mathbb{E}_{\pi_S}$ and $\mathbb{E}_{\pi_R}$ is one of our contributions. Specifically, $C_t$ in $\hat{X}_t$ in $\pi_{Rt}(a_{Rt}|\hat{X}_t)$ is generated by the differentiable renderer $G$ with the actions $a_{St}$ from the sender policy $\pi_S(a_{St}|I_S, C_{t-1})$. Hence, we can have both $\partial \mathcal{V}/\partial \pi_{Rt}$ and $\partial \mathcal{V}/\partial \pi_{St}$ based on Eq. (5). This results in a novel multi-agent variant of the general policy gradient (Sutton et al., 2000; Silver et al., 2014).

$V_\lambda(X_t)$ in Eq. (5) is an eligibility trace approximation (Sutton and Barto, 2018) of the ground-truth value function. Intuitively, a value estimate with eligibility trace $V_\lambda$ mixes the bootstrapped Monte Carlo estimate $V_N^k(X_t) = \mathbb{E}_{\pi_S, \pi_R}[\sum_{n=t}^{h-1} \gamma^{n-t} r_n + \gamma^{h-t} \delta(a_{Rh}) v_\phi(X_h)]$ at different roll-out lengths $k$, with $h = min(t + k, T_{choice})$ being the maximal timestep. $T_{choice}$ is the timestamp when the receiver stops waiting. The articulation of such termination also makes our eligibility trace deviate from the general derivation with infinite horizon. We derive an episodic version as

$$V_\lambda(X_t) = \begin{cases} (1-\lambda) \sum_{n=1}^{H-1} \lambda^{n-1} V_N^n(X_t) + \lambda^{H-1} V_N^H(X_t) & \text{if } t \leq T_{choice} \\ v_\phi(X_t) & \text{otherwise,} \end{cases} \quad (6)$$

where $H = T_{choice} - t + 1$. Please refer to Appendix C for a detailed derivation. Finally, $v_\phi(X_t)$ is trained by regressing the value returns:

$$\phi^* = \arg\max_\phi \mathbb{E}_{\pi_S, \pi_R}\left[\sum_t \frac{1}{2}\|v_\phi(X_t) - V_\lambda(X_t)\|^2\right]. \quad (7)$$

The training algorithm is summarized in Alg. 1.

## 5 Experiments

### 5.1 Settings

**Images** We used the Sketchy dataset (Sangkloy et al., 2016) as the image source. Due to the limited performance of the sketching module on open-domain image-to-sketch sequential generation, we select 40 categories (10 images per category, see Appendix A) with satisfactory sketching behaviors.

**Environmental drivers** With the visual communication game and the learning agents at hand, we investigate the factors in emergent graphical conventions with controlled experiments. Tab. 1 lists all designed settings. Specifically, we consider the following:

---

**Algorithm 1: Training Algorithm**

**Initialization :** Initialize neural network parameters $\theta$, $\rho$, $\phi$ for $\pi_S$, $\pi_R$, and $v_\phi$, respectively.

1 **for** *game round $l = 1, ..., L$* **do**
2     **for** *time step $t = 0, ..., T$* **do**
3        $a_{St} \sim \pi_S(a_{St}|C_t, I_S)$
4        $C_{t+1} = G(C_t, a_{St})$
5        $a_{Rt+1} \sim \pi_R(a_{Rt+1}|C_t, C_{t+1}, I_R^1, ..., I_R^M)$
6        **if** *$a_{Rt+1}$ is not wait* **then**
7           $T_{choice} = t$
8        **end**
9     **end**
10     Compute $\{V_\lambda(X_t)\}_{t=1}^T$ via Eq. (6)
11     Compute $\{\mathcal{V}(X_t)\}_{t=1}^T$ via Eq. (5)
12     Update $\theta \leftarrow \theta + \alpha_S \nabla_\theta \sum_t \mathcal{V}(X_t)$
13     Update $\rho \leftarrow \rho + \alpha_R \nabla_\rho \sum_t \mathcal{V}(X_t)$
14     Update
       $\phi \leftarrow \phi - \alpha_v \nabla_\phi \sum_t \frac{1}{2}||v_\phi(X_t) - V_\lambda(X_t)||^2$
15 **end**

---

Table 1: **Game settings and results.** The first three columns represent the configurations of the environmental drivers. Setting names and descriptions specify our purposes for intervention. The last three columns show success rates and conversation length in testing games. Games marked with "seen" are validation games with the same image set as training. Games marked with "unseen" are testing games with unseen images.

| Game Settings | | | | | Communication Accuracy (%) $\pm$ SD (avg. step) | | |
|---|---|---|---|---|---|---|---|
| early decision | update sender | max/one step | description | setting names | seen | unseen instance | unseen class |
| yes | yes | max | our experimental setting | complete | $98.07 \pm 0.01(1.03)$ | $70.37 \pm 0.04(2.36)$ | $39.40 \pm 0.05(3.76)$ |
| no | yes | max | control setting for early decision | max-step | $86.27 \pm 0.03(7.00)$ | $67.93 \pm 0.02(7.00)$ | $38.40 \pm 0.04(7.00)$ |
| yes | no | max | control setting for evolving sender | sender-fixed | $99.60 \pm 0.01(2.41)$ | $71.80 \pm 0.02(3.83)$ | $45.40 \pm 0.02(4.75)$ |
| yes | yes | one | control setting for sequential game | one-step | $22.87 \pm 0.23(1.00)$ | $14.07 \pm 0.15(1.00)$ | $9.60 \pm 0.09(1.00)$ |
| no | no | max | baseline for all settings above | retrieve | $99.47 \pm 0.01(7.00)$ | $76.80 \pm 0.02(7.00)$ | $48.00 \pm 0.02(7.00)$ |

- *Can receiver make early decisions?* The hypothesis is that the receiver's decision before exhausting the allowed communication steps may inform the sender of the marginal benefit of each stroke and incentivize it to prioritize the most informative strokes. The corresponding control setting is *max-step*, wherein the receiver can only make the choice after the sender finishes its drawing at the maximum step. This factor is on in other settings.

- *Can the sender change its way of drawing?* The hypothesis is that the mutual adaptation of the sender and the receiver may lead to better abstraction in the evolved sketches. Particularly, the sender may develop new ways of drawing in the evolution. The corresponding control setting is *sender-fixed*, wherein we freeze the parameters of the sender such that the receiver has to adapt to its partner. This factor is on in other settings.

- *Is the game sequential, and can the receiver observe more complex drawings?* The hypothesis is that the modeling of a sequential decision-making game and the evolution from more complex sketches may regularize the arbitrariness, which is unique for graphical conventions. The corresponding control setting is *one-step*: There only exists one step of sketching, thus no sequential decision-making at all. This factor is on in other settings.

**Training, validation, and generalization**   We train the sender/receiver with batched forward and back-propagation, with a batch size of 64 and maximum roll-out step $T = 7$. We update using Adam (Kingma and Ba, 2015) with the learning rate 0.0001 for a total of 30k iterations. We train each model on a single Nvidia RTX A6000; one experiment takes 20 hours. Except for the *sender-fixed* setting, there is a warm-up phase in the first 2000 iterations for the sender where we linearly increase the learning rate to 0.0001. After the warm-up phase, the learning rate of both agents will be decreased exponentially by $0.99^{\frac{i-2000}{1000}}$, where $i$ is the number of training iterations. In all settings, we set $M = 3, \gamma = 0.85$. Between iterations, we randomly sample another 10 batches for validation. We use 30 categories (10 images per category) for training and held out 10 images per category for the unseen-instance test; another 10 categories are for the unseen-class test. Each image is communicated as the target, resulting in 300+100 pairs in the test set. At each iteration, the categories and instances are sampled randomly to constitute the context. Results henceforth are statistics of 5 random seeds.

## 5.2   Results

### 5.2.1   Communication efficacy and sketch abstraction

We record both the success rate and the communication steps over the training iterations; see Fig. 3. The communication is considered successful when the receiver makes the correct prediction at the end of the game. In Fig. 3a, agents in all settings except *one-step* converge to a success rate above 80%. Among them, the communicating pairs in the *complete* setting and the *sender-fixed* setting evolve to achieve a comparable success rate with the *retrieve* baseline. Interestingly, these two pairs also emerge a phenomenon resembling the natural observation in human studies, named *systematic reduction* (Lewis, 1969): The average steps first increase and then gradually decrease as in Fig. 3b. Contrasting *complete* and *sender-fixed*, we can see: (i) The emergent conventions in the former is much simpler than the latter (less steps in Fig. 3b), which implies the contribution of mutual adaptation in sketch abstraction. (ii) The success rate in Fig. 3a in the former converges a

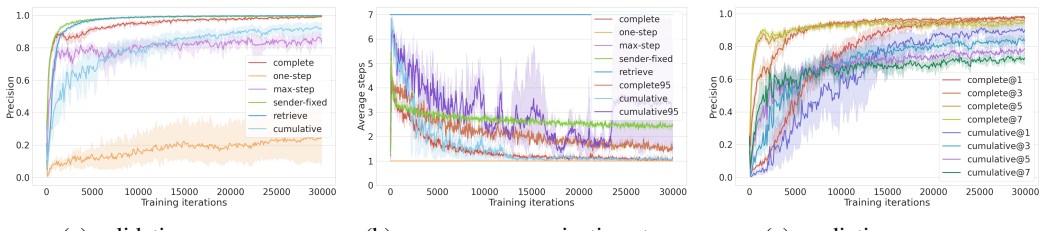

| (a) validation accuracy | (b) average communication steps | (c) prediction accuracy |

Figure 3: **Statistics aggregated over all random seeds.** (a) The validation accuracy of different game settings and the ablation baseline. (b) The average communication steps under different settings and ablation baselines. $\gamma$ is 0.85 by default, and 0.95 in complete95 and cumulative95. (c) The prediction accuracy when receivers are presented with sketches drawn by corresponding senders at 1, 3, 5, and 7 steps, respectively. These sketches are collected in a standalone roll-out after each iteration, where the early decision is disabled; agents are trained with the *complete* setting, where the early decision is enabled.

bit more slowly, which is reasonable since the senders can explore and change the way of drawing. In comparison, if the receiver cannot make early decisions, it has no intention to relate sketches (*i.e.*, $C_{t-1}$ and $C_t$) at consecutive timesteps. The sender is thus *unaware* of each stroke's marginal information gain, which in return makes their learning harder. This might explain the relatively low success rate in the *max-step* setting. The failure of the *one-step* pairs reveals the irreplaceable roles of sequential decision-making and observing complex sketches in emergent graphical communication.

To further inspect how our proposed modeling and learning on sequential decision-making facilitate the desired evolution in the sketches, we conduct an ablation study by comparing our proposed learning surrogate (Eq. (5)) and a vanilla policy gradient baseline, REINFORCE (Williams, 1992) with Monte Carlo cumulative rewards $\mathbb{E}_{\pi_S, \pi_R} \left[ \sum_t \left[ \nabla \log \pi_R(a_{Rt}|\hat{X}_t) \sum_{n=t}^{T} \gamma^{n-t} r_n \right] \right]$.

Our comparison spans three axes. First, the REINFORCE baseline converges much more slowly than the proposed surrogate; see *cumulative* vs *complete* in Fig. 3a. Second, we check the robustness under the variation of decay factor $\gamma$. As shown in Fig. 3b, while the proposed method shows stable convergence in the communication steps under $\gamma = 0.85$ and $0.95$, the REINFORCE baseline exhibits high-variance behavior under certain setup ($\gamma = 0.95$). Third, we check if agents' early terminations are *caused* by their *awareness* of the indistinguishable performance in longer and shorter episodes. Given a *precondition* that the longer the episodes are, the earlier the success rate increases, it should be the increase in the average performance of shorter episodes that *causes* the average timesteps to decrease. Taking 1-step and 3-step communication for example, in the *complete* setting, we shall see the success rate of the 3-step to achieve high earlier, which is then caught up but not exceeded by the 1-step. The not exceeding condition is a crucial cue to validate that the communicating partners were *actively* pursuing the Pareto front (Kim and De Weck, 2005) / efficiency bound (Zaslavsky et al., 2018) of accuracy and complexity. This is exactly what our proposed method emerges as shown in Fig. 3c. In contrast, in the REINFORCE baseline, under the same decay factor, the performance of 1-step surpasses 3-step communication. It seems as if the incapability of *learning* long episodes *caused* the agents to *avoid* them.

Together, all our results on success rate and communication steps are consistent with predictions based on our hypotheses, which justifies our environments and models. However, a high success rate does not necessarily imply high convention quality. Another essential property for conventions is *stability* (Lewis, 1969): There should exist common patterns in the repeated usage of conventions to convey similar concepts. We take the viewpoint of representation learning and concretize the vague *stability* with three formally defined properties: *iconicity*, *symbolicity*, and *semanticity*. Below, we introduce our experiments to measure these properties systematically.

### 5.2.2 Iconicity: generalizing to unseen images

We define *iconicity* as the drawings exhibiting a high visual resemblance with the corresponding images, such that they are proximal to the latter when measured on the high-level embedding of a general-purpose visual system. To quantitatively measure the visual similarities, we need to compute the distance between the sketch and image in the visual embedding space (*e.g.*, with cosine similarity). However, since we do not know how the embeddings distribute in their space, this unnormalized measure is prone to model-specific biases. Fortunately, the receiver's policy takes the form of a softmax of the cosine similarity between the embedding of the sketch and a context set with a target image and some randomly sampled distractors, which naturally approximates the normalization we want. Therefore, communication accuracy can serve as a visual similarity measure, and its *generalizability* to unseen images can pinpoint the emergent preservation of *iconicity*. Intuitively, in open-domain communication, agents would see novel scenes with known or unknown concepts— unseen instances of seen classes and unseen classes, respectively. They should still be able to communicate with established conventions or with unconventionalized iconic drawings. A successful generalization to unseen classes (*i.e.*, zero-shot generalization) is more difficult than unseen instances; partners may increase the conversation length and communicate with more complex sketches.

Tab. 1 reports the success rates and average timesteps in our generalization tests. The *retrieve* setting is the baseline since there is no evolution at all and the sketches should resemble the original images the most (*i.e.*, possessing the highest *iconicity*). Unsurprisingly, its generalization performance upperbounds all other settings. Among the experimental and controlled settings, the *complete*, the *max-step*, and the *sender-fixed* agents generalize relatively well in unseen instances ($70.37 \pm 0.04$, $67.93 \pm 0.02$, $71.80 \pm 0.02$) and generalize above chance ($39.40 \pm 0.05$, $38.40 \pm 0.04$, $45.40 \pm 0.02 > 25.00$) in

unseen classes. Interestingly, *complete* and *sender-fixed* communicating partners intelligently turn to longer episodes for better generalization, better than the *max-step* agents. This finding implies the partners may turn to more iconic communication when there is no established conventions/symbols, just as we humans do. Strikingly, the *max-step* conventions seem to loose more *iconicity*, possibly due to confusion on marginal information gains. The *one-step* drawings seem to lack *iconicity*.

### 5.2.3 Symbolicity: separating evolved sketches

Next, we measure *symbolicity* to evaluate the graphical conventionalization. We define *symbolicity* as the drawings being consistently separable in the high-level visual embedding, which facilitates new communication partners to easily distinguish between categories without grounding them to referents. Based on this definition, a more *symbolic* drawing should be more *easily separable* into their corresponding categories. To measure such *separability*, we use a pre-trained VGG16 as the new learner and finetune the last fully connected layer to classify evolved sketches into the 30 categories. Technically, we first get the 300 final canvases from the communication game, 10 for each category. Among them, we use 70% for training and 30% for testing.

The bar plot in Fig. 4 shows the classification results. Since agents in the *one-step* setting do not play the game successfully, they may not form a consistent way to communicate. Agents in the *complete* setting achieve the highest accuracy, higher even compared with the result of the original images. This finding indicates that agents in the *complete* setting agree on a graphical convention that consistently highlights some features across all training instances in each category, which are also distinguishable between categories. Comparing the *max-step* with the *complete* setting, we know that early decision is a crucial factor for more *symbolic* conventions. Comparison between the *sender-fixed* setting and the *complete* setting suggests that the sender's evolution also contributes to high *symbolicity*.

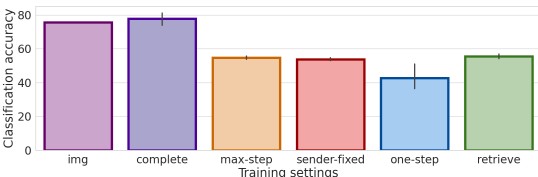

Figure 4: **Testing results of classifiers trained with sketches evolved under different settings.** *img* denotes images, and *retrieve* denotes unevolved sketches.

Table 2: **Semantic correlation between the word vector space of the communicated target and the feature vector space of the trained VGG.**

| setting | correlation |
|---|---|
| complete | $0.43 \pm 0.02$ |
| max-step | $0.41 \pm 0.13$ |
| sender-fixed | $0.36 \pm 0.06$ |
| one-step | $0.31 \pm 0.08$ |
| retrieve | $0.55 \pm 0.00$ |

### 5.2.4 Semanticity: correlating category embedding

The last desired property is that the evolved sketches can preserve the perceptual metric in images (Zhang et al., 2018). We define *semanticity* as the topography of the high-level visual embedding space of drawings being strongly correlated to that of images, such that semantically similar instances and categories lie close to each other in the embedding space. To examine such *topographic correlation*, we project category names to the feature space using word2vec (Mikolov et al., 2013) as featureA, and project the evolved sketches to the feature space using the trained VGG in Sec. 5.2.3 as featureB. We compute the correlation of the distances between all the possible pairs of featureB and the corresponding pairs of featureA as the semanticity measure. The results in Tab. 2 show that semanticity

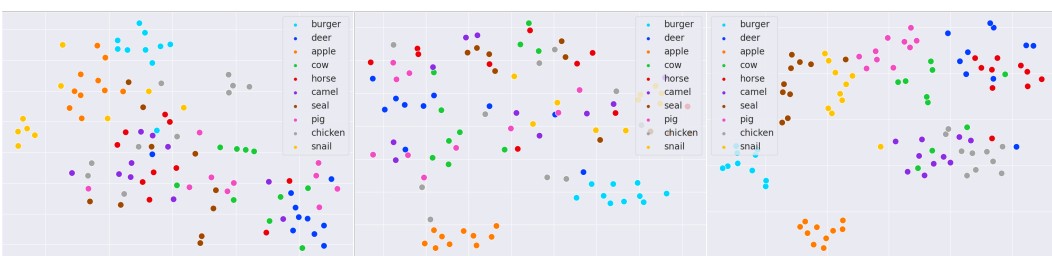

Figure 5: **t-SNE of visual embedding of the original images (left), unevolved sketches in the *retrieve* setting (middle), and evolved sketches in the *complete* setting (right).** These embeddings are from the finetuned VGGNets in Sec. 5.2.3. The evolved sketches have clearer boundaries due to the discrimination game, while maintaining the topography such that similar concepts are close to each other.

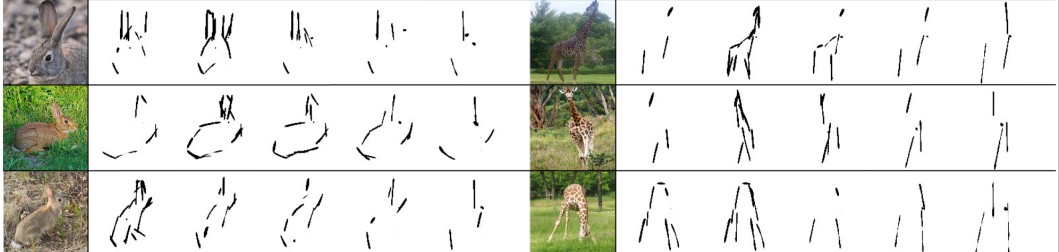

Figure 6: **Sketch evolution of rabbit and giraffe.** For each example, sketches from the left to the right show the change of the final-step canvas from iteration 0 to 30,000. Please refer to Appendix D for more results.

can be better retained in the *complete* setting compared with the *retrieve* baseline. We further visualize the embeddings of the sketches in a 2D space via t-SNE (Van der Maaten and Hinton, 2008). Fig. 5 show the visualization of the original images and the sketches in the *retrieve* and *complete* settings; please refer to Appendix B for results of other settings. Images/drawings from the same category are marked in the same color. As shown, boundaries between categories are clearer in the evolved drawings in the *complete* setting than in *retrieve* or original images; but semantically similar categories are still close to each other. For example, cow, deer, horse, and camel are proximal to each other, while burger and apple are far from them. These results highlight another uniqueness of visual communication over its symbolic counterpart: The similarity in the visual cues in the conventions may hint the *semantic* correlation between the referents.

### 5.2.5 Visualizing sketch evolution

To better understand the nature of the emerged conventions, we inspect the intermediate sketches in the evolution processes. Specifically, we visualize the process under the *complete* setting. Fig. 6 shows three instances in two categories. For each example, drawings from the left to the right show the change of the final-step canvas. Sketches' complexity gradually decreases after an initial increase, echoing the trend of reduction described in Sec. 5.2.1. For rabbits, in the beginning, the strokes may depict instances from different perspectives; through iterations, they converge to highlight the rabbit's long ear. As for the giraffe, the agents gradually learn to emphasize the long neck. In the third example, although the giraffe lowers its head, we can still see an exaggerated vertical stroke for the neck, similar to the first example where the giraffe's head is raised. These examples show how the sender gradually unifies drawings of the same category: After evolution, the sender is inclined to use the first several strokes to depict the most salient parts of visual concepts.

## 6    Conclusion

We present the first step of modeling and simulating the evolution of graphical conventions between two agents in a visual communication game. Agents modeled in our framework can successfully communicate visual concepts using sketches as the medium. We measure the emergent graphical conventions over three carefully defined properties, *iconicity*, *symbolicty*, and *semanticity*. The experimental results under different controls suggest that early decision, mutual adaptation, and sequential decision-making can encourage *symbolicity* while preserving *iconicity* and *semanticity*. However, two-stage pre-trained senders possess limitations; an ideal sender would not need to convert the images to pixel-level sketches before it starts sketching. The limitations in the pre-trained sketching module also constrain the discriminative need among the selected classes. We hope to investigate these limitations in future research. With the uniqueness of visual conventions demonstrated, we hope our work can invoke the study of emergent communication in the modality of sketches.

**Acknowledgments**    J. Joo was supported in part by the NSF SMA/SBE #1831848. We would like to thank Prof. Marco Baroni (Pompeu Fabra Univ.) and Prof. Yaodong Yang (Peking Univ.) for helpful comments on the draft, Miss Chen Zhen (BIGAI) for making the nice figures, all anonymous reviewers for their constructive comments on prior submissions, and Prof. Ying Nian Wu (UCLA), Prof. Hongjing Lu (UCLA), Prof. Federico Rossano (UCSD), Dr. Zilong Zheng (BIGAI), Dr. Chi Zhang (BIGAI), Baoxiong Jia (UCLA), Dr. Qing Li (BIGAI), and Dr. Bo Pang (UCLA) for discussion on the algorithm and experimental designs.

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
