# OpenReview forum: "Emergent Graphical Conventions in a Visual Communication Game"
_NeurIPS.cc/2022/Conference — NeurIPS 2022 Accept_

### Official Review · Reviewer_b2Pi · 2022-07-10

**Rating:** 6
**Confidence:** 5
**Soundness:** 3 good
**Presentation:** 3 good
**Contribution:** 3 good

**Summary:**

The authors proposed to address the problem of visual communication with graphical sketches. By proposing a reinforcement learning framework, two agents (a sender and a receiver) play a visual communication game to reach an agreement on what the visual message is. There are three key factors used to provide some insights inside the visual communication. Experiments are conducted on a subset of the sketch-photo paired dataset, i.e., Sketchy, to validate the proposed method. Some interesting results are provided as well.

**Questions:**

(1) Did the authors try to use alternative sketching methods to measure the impact of sketching quality? One possible candidate could be CLIPasso [A].

[A] CLIPasso: Semantically-Aware Object Sketching

(2) At each time step, five new strokes will be added on the previous canvas $C_{t-1}$. Is there an action to remove existing strokes? I personally think some previously drawn strokes will become redundancy when the sender evolves.

(3) Human study on sketch evolution would be helpful to confirm the quality of the generated sketch by the sender.

(4) Is that the final sketching could be somehow matched to the most informative strokes to represent the object in the original natural image? This could be interesting to know.

**Ethics Review Area:**

["I don’t know"]

**Limitations:**

The authors have discussed the limitations in terms of two-stage pretrained senders.

**Strengths And Weaknesses:**

Strengths
- This works proposed to address a significant problem of emergent graphical conventions by designing a visual communication game using sketches, which is very interesting and inspiring.  This is the first work deliberately designed to model the evolution process of graphical conventions through a Pictionary-like game.
- The technical contributions are concrete. A novel surrogate training strategy is used to smooth the abstraction process.
- Iconicity, symbolicity and semanticity are proposed to evaluate the emergent graphical conventions, which are of great importance to enable quantitative measurements on visual communication.

Weaknesses
- The effectiveness of the communication process might be influenced by the quality of the target sketch $\hat{I_S}$ produced from image $I_S$.
- The data used for training and testing is very small, i.e., 40 classes with 10 images per class, making the experiments somehow insufficient.

---

> ### Author Response · Authors · 2022-08-02
> **Thank you for your review**
>
> Thanks for all your comments and for acknowledging our work studying an interesting problem and proposing important evaluation measures.
>
> >**[Q1]:** The effectiveness of the communication process might be influenced by the quality of $\hat{I_S}$.
>
> The current design of the sender indeed bottlenecks the performance of the evolved agents and limits our dataset size. Thanks for your suggestion. The results of CLIPasso are quite impressive. We will explore the model and modify it into a sequential version so it can be adopted into our game. When more data is added, further experiments varying the number of categories and the number of distractor images can add new facets to this topic.
>
> >**[Q2]:** Action to remove existing strokes.
>
> We do not consider the action to remove the existing strokes. Our training objectives indicate whether the communication is successful or not and the temporal cost encourages a short conversation length. The sender is supposed to learn to draw fewer redundant strokes through the iterative training process.
>
> >**[Q3]:** Human study on sketch evolution.
>
> Thank you for your suggestion. We will include it in future work.
>
> >**[Q4]:** Do the final sketches matched the most informative strokes?
>
> From our observation, the senders do evolve to use the first strokes to highlight the most salient part of the object, as shown in the final sketches of the examples in Fig 6 and other visualization results in the supplementary material.

---

### Official Review · Reviewer_6BRM · 2022-07-11

**Rating:** 6
**Confidence:** 5
**Soundness:** 2 fair
**Presentation:** 1 poor
**Contribution:** 2 fair

**Summary:**

This paper explores the modality of sketches in the context of emergent communication games with visual referents. While other papers have already studied graphical channels as an alternative to discretised tokens explored in prior emergent communication research (e.g. ref 29 in the paper), the unique feature of this work is studying the evolution of these conventions in a multi-step game. The authors define three properties (iconicity, symbolicity and semanticity) to study the evolved sketches with quantitative and qualitative methods. Results show that the agents are able in most cases to learn successful communication channels.

**Questions:**

- Can you clarify that M=3 (L231) you mean that there are only three context images? If so,
    - Why is this set so low? Other works with referential image games have used significantly larger distractor sets and still achieved good to near perfect communication (both with sketching and with token based messages). Further, the unseen images seem to behave significantly worse, whereas in many emergent communication studies performance on unseen test data is often in line with the training data performance - what causes this?
    - How is it that the communication accuracy is so low in some settings? In the one-step setting it would be significantly worse than random guessing; for many of the other "unseen class" settings it is barely better than random
- What happens when agents from pairs of different communication scenarios are tested (i.e. different image subsets)? Are the conventions emergent from disjoint subsets still holding up?
- The proposed measure of semanticity is only demonstrated qualitatively; could a quantative measure not be developed? It also would be interesting to understand how this relates to design choices in other works which explictly try to enforce this (e.g. the "perceptual" losses in 29 and also in Mihai, D and Hare, J, "Shared Visual Representations of Drawing for Communication: How do different biases affect human interpretability and intent?", SVRHM 2021)


**Limitations:**

As it stands whilst the authors try to draw attention to some of the shortcomings of their work, they do it out of context with respect to the published literature. As highlighted above in the the questions and the weaknesses there are some particular areas where limitations could be better drawn-out and discussed.

**Strengths And Weaknesses:**

## Strengths

- Studying emergent communication through the modality of sketches is an exciting direction, although this paper is not the first to explore it (paper ref 29, but also see Fernando et al, 2020, From language games to drawing games, arXiv preprint arXiv:2010.02820). The study of the *evolution process* of the emergent sketches between two cooperative artificial agents, however, is a fruitful initiative to be explored and the authors compare it to observations from human studies.
- Using the previous canvas as an external memory and comparing against it, among other distracting referents, to incentivise the sender to draw informative sketches as early on in the communication is a nice idea.
- This work proposes three evaluation metrics to assess the evolved sketches; these are a valuable contribution for the field going forward.

## Weaknesses

- This work, in its current form, doesn't take into account the current state of emergent graphical communication and advances in differentiable drawing made in the last two years. These could have helped address some of the problems highlighted in this work (e.g. the sender first using a visual module that turns a natural image into a pixel sketch is something overcome by many recent works).
- Building upon the above, the comparison against other similar models is somewhat lacking; for example the category oriented game setting is seemingly the same as the "object-oriented" setting explored by ref 29 (which was published at NeurIPS last year).
- Due to the choice of sketching module, only 40 categories (with only 10 images per category) of Sketchy dataset are explored in this paper. Other works have considered much more diverse image sets and considerably harder games.
- Communication performance seems to be relatively low compared to other approaches given the simplicity of games with only two distractors/context images. It's not clear why this is.
- The new measures proposed are mostly only evaluated qualitatively.
- Human performance against the agents is not considered as it has been in related works.


# Other remarks

- L17 of the abstract "the unexplored modality of sketches" isn't really true. Sketching has already been looked at by other papers. It would be much better to refocus the paper on the contribution of this work in studying the evolution of graphical communication and providing new measures. Similar issues appear on L97, L103 and L366.
- Section 2 "Learning to sketch" does not give an accurate picture of the state of the art in this space; there is considerable work in that goes well beyond "the early phase". For example:
    + A. Das et al, 2021. Cloud2curve: Generation and vectorization of parametric sketches. CVPR 2021
    + T.Z. Li et al. Differentiable Vector Graphics Rasterization for Editing and Learning. SIGGRAPH Asia 2020
    + D. Mihai and J. Hare, 2021. Differentiable Drawing and Sketching. arXiv:2103.16194
    + D. Smirnov et al, 2020. Deep parametric shape predictions using distance fields. CVPR 2020
    + H. Mo et al, 2021. General virtual sketching framework for vector line art. ACM Transaction on Graphics 2021
- Section 5.2.5 would benefit from expansion and more discussion; there are contributions here.
- Some of the refs need updating to the published papers rather than arxiv versions (e.g. 23, 29)
- Please define communication accuracy for the benefit of readers not familiar with emergent communication games

---

> ### Author Response · Authors · 2022-08-02
> **Thank you for your review[1/2]**
>
> Thanks for your thorough reading and all your comments.
>
> >**[Q1]:** SOTA of emergent graphical communication.
>
> We want to clarify that we did **not** claim we are the first to "learn to sketch through communication," the credit of which we believe shall certainly go to [29]. Instead, we phrased our first contribution as "the first step to modeling and simulating the evolution process of graphical conventions." In L26-31, we explicitly disperse the concepts of "graphical conventions" and "sketches" along the axis of "iconicity" vs "symbolicity." Since our focus is "graphical conventions," our model should be evaluated independently from how the agents learn to sketch before entering the communication game. Works sharing similar ideas and settings are compared in "Emergent graphical conventions" in Section 2.
>
> Thanks for your comment, we added the missing citation and update the citations to the published version in the revised version. We also changed our statements as mentioned in L17, 97, 103, and 366(L17, 82, 106, 371 in the revised version).
>
> >**[Q2]:** SOTA of learning to sketch and differentiable drawing.
>
> In Section 2 "learning to sketch", we mainly talks about work focusing on **image** based on sketch generation from **various** categories of natural image. Since we want to simulate the process of how the sketch evolves from **high iconic** to a more abstract drawing trading-off between accuracy and complexity in a **sequential** game, the sender is required to (1) draw the sketch closely resembling the given a target image from a **random** class, and (2) draw strokes sequentially and the order of the strokes can be readjusted later.
>
> Thanks for all the drawer models mentioned in your comment. A. Das et al, 2021 focused on sketch to curve generation. T.Z Li et al and D. Mihai and J. Hare, 2021 proposed ideas for differentiable drawing. But they still need a sketch-like image input (from $I_S$ to $\hat{I_S}$) and cannot be directly applied to image-based **sequential** sketch generation. D. Smirnov et al, 2020 targets at sythentic images of rigid shapes, for example letter A, chair, etc. However, our input are natural images. The line drawing of faces in H. Mo et al, 2021 shows promising results but cannot be directly applied to more than 10 classes. Taking all these into account, image-based sketch generation of open-ended classes seems to still be a challenging task, especially when the sketch is required to be generated sequentially. Due to page limit, we will add more related work to this section in the future version.
>
> The pretraining of the sender indeed bottlenecks the performance of the evolved agents. Therefore, (1) we have a setting called retrieve that served as an unevolved baseline. Contrasting complete and retrieve, we observed our agents do form "graphical conventions" without sacrificing too much iconicity. (2) In future, we will explore the work mentioned by reviewer b2pi and [29] in a sequential setting to alleviate the influence of the pretrained sender.
>
> >**[Q3]:** Game setting seemingly the same as the "object-oriented" setting in [29]?
>
> Context design is only a part of the communication game. Even though the category-level context was introduced in prior work such as [26, 29], our game is different from [26] with sketch as the communication medium and different from [29] due to its sequential turn-taking nature. We recognized the contribution in context design from these prior works at L125 in the revised version.
>
> >**[Q4]:** Small context size?
>
> Due to the sequential nature of our game, the training can be computationally expensive. Therefore, we keep the game size small and also propose a novel multi-agent reinforcement learning strategy to make the training process more effective.
>
> In this work, we highlight experiments on major confounding factors to validate the proposed model, the training framework, the definition of graphical convention properties, and their corresponding measures. Further experiments can be added by varying the number of categories, the number of distractor images, the distribution of the context images, and the number of images in each category to explore the influence of other environmental factors.
>
> >**[Q5]:** Low performance of one-step and the unseen test class?
>
> Please note the receiver is not forced to choose at the end of the game. If the task is difficult (sketches are not informative in the one-step setting; the agents are not familiar with the referents based on their built convention), the receiver will choose to wait for further drawings. The increase in the conversation length in the unseen test class in table 1 also manifests this phenomenon.

---

> > ### Author Response · Authors · 2022-08-02
> > **Thank you for your review[2/2]**
> >
> > >**[Q6]:** What happens when agents from pairs of different communication scenarios are tested (i.e. different image subsets)?
> >
> > Due to the time limit, we only trained three pairs of agents under the complete setting. Each pair only shares 5 images per category with other pairs, with the remaining 5 images private to them. These private subsets are disjoint from each other. We test the communication accuracy on disjoint subsets that are unseen during training. As shown below, such accuracy is consistent between pairs. It is also consistent with the unseen-instance generalization results reported in table 1.
> >
> > | setting     | disjoint subset|
> > |-------------|-----------------|
> > |pair A| 72.81
> > |pair B| 70.73
> > |pair C| 69.90
> >
> > >**[Q7]:** Quantitative evaluation of semanticity? Design losses to enforce the emerged properties?
> >
> > We project the category names to the feature space using word2vec [30] as featureA, and project the evolved sketches to the feature space using the trained VGG in Sec 5.2.3 as featureB. We compute the correlation of the distances between all the possible pairs of featureB and the corresponding pairs of featureA as the semanticity measure. The results shown below is consistent with our original results: semanticity can be better retained in the complete setting.
> > | setting     | correlation |
> > |-------------|-----------------|
> > |complte|0.43 $\pm$ 0.02|
> > |max-step|0.41 $\pm$ 0.13|
> > |sender-fixed|0.36 $\pm$ 0.06|
> > |one-step|0.31 $\pm$ 0.08|
> > |retrieve|0.55 $\pm$ 0.00|
> >
> > Different from "learning to draw" from scratch, the essential idea of “emergence” is that some properties naturally come into being without being explicitly encouraged to do so. Please note in Fig. 5 and the results shown above that our observation is not that sketches evolve to become "semantic"; the unevolved sketches are already somewhat "semantic," and the evolution **preserves** such semanticity without explicit guidance. Semanticity is the property we want to evaluate when new graphical conventions spontaneously emerged from different environments. We did not design designated terms within the model, the game objective, or the training surrogate to direct agents’ evolution.
> >
> > >**[Q8]:** Clarification on accuracy and more discussion on Sec 5.2.5.
> >
> > Thanks for your suggestion. We have added the definition of communication success in L244 in the revised version. Due to page limit, we will add more discussion on Sec 5.2.5 in future version.

---

> > > ### Comment · Reviewer_6BRM · 2022-08-09
> > > **Thank you**
> > >
> > > Thank you for the response and updates; I'd be happy to increase my score on the basis of these revisions and clarifications being made in the paper.

---

### Official Review · Reviewer_kX83 · 2022-07-16

**Rating:** 6
**Confidence:** 3
**Soundness:** 3 good
**Presentation:** 3 good
**Contribution:** 3 good

**Summary:**

This work is an attempt at mimicking communication processes in humans that predate language, namely the process of drawing sketches. Here, a sender tries to draw an image via minimalist sketches step by step in an attempt to communicate with a receiver. The work claims a new value function based MARL training objective and defines three aspects of communication via the sketches: iconicity, symbolicity, semanticity. Results align with the author's expectations of the evolution of emergenet communication.


**Questions:**

How'd you end up with the learning rate schedule and other parameter that you did, they seem relatively non standard (perhaps the result of a grid search?)? - lines 224-236


**Limitations:**

Authors discussed limitations but limited themselves to the accuracy of the vision systems available (the need to convert from real to pixellated images, etc.). What could be better discussed as limitations is the ability to evaluate along the three dimensions proposed in the work without the aid of human annotators.

**Strengths And Weaknesses:**

*Strengths:*

- The paper is quite refreshing in its motivation: study emergent communication using sketches (as opposed to much of the current literature which uses discrete token based communication).

- This line of work could open up ways to study emergence of pictographic languages, a quite underexplored region of the emergent communication space.

- I quite like the introduced measures of iconicity, symbolicity, semanticity - their underlying definitions are exactly what is needed to measure effectiveness of the communication in such referential games.


*Weaknesses:*

- The clarity of the novelty surrounding the training objective via joint value function is unclear with respect to prior works.

- The results surrounding how the sketches evolve into semantically similar classes is interesting and would serve to validate the motivation of this paper, but is rather difficult claim to assess the veracity of. As it appears to be a side effect of the proposed training objective, should the take away be that this particular objective better models emergent communication through sketches in humans?

- A summarized sentence on why the observation spaces between sender and receiver are disjoint (line 120) would be useful instead of just a citation. (The images themselves are distinct but can belong to the same partition of class is an additional missing clarification.)

- Fig 1 choices might not be the best with respect to conveying the goals of the work. Line 170-171 clearly say that the receiver only picks an image when they have info to distinguish from the distractors. The round 1 of Fig 1 already appears to have more than sufficient information to distinguish rooster from the other two, it is only when it is simplified in round 2 that there is ambiguity with the other bird. This calls into question how ''good`` the distrctor images are and whether most of the games can just be solved with maximal reward for both agents if sketching abilities were high within a single step.
    -  A side effect of this is that it is not clear whether the emergent communication happens because of the communication aspect or if it is simply dependent on the actual drawing abilities of the sender.

- The claims regarding symbolicity and semanticity seem viable to judge through automated methods as in the paper but I am not clear how this could work for iconicity. High visual resemblance to the images is derived from the perspective of an agent - either a human or one of the learning agents based on their training objective. Without such a reference to ground these numbers, once again it is difficult to assess the accuracy of the lines 359-361.

---

> ### Author Response · Authors · 2022-08-02
> **Thank you for your review[1/2]**
>
> Thank you for all your valuable feedback and for recognizing the impact and the proposed evaluation measures.
>
> >**[Q1]:** Clarification of the novelty of the training method.
>
> We believe our training method is novel because agents are learned with policy gradients backpropagated through differentiable transition models from a training surrogate that combines value function approximation and eligibility trace estimation. The value function is introduced to reduce variance and the eligibility trace can correct bias [39]. Prior works that emerges symbolic languages in sequential communication with early termination [3,6] uses REINFORCE to derive the training surrogate. In a comparison with REINFORCE in L258-272(L263-277 in revised version), we show that our surrogate can effectively smooth the training process and help the agents correlate between complex and simple sketches while the REINFORCE agents struggle with longer conversations. In the broader literature of multi-agent turn-taking games, each agent is modeled with its own value function, the TD learning of which is challenging without modeling other agents' policies [1]. In comparison, our training surrogate (1) leverages eligibility trace, i.e., a mixture of Monte Carlo estimate at different roll-out lengths, bypassing the challenges of turn-taking based TD learning, and (2) organically incorporate the sender's and the receiver's policies for policy gradient update.
>
> [1] Wen, Ying, et al. "Probabilistic recursive reasoning for multi-agent reinforcement learning." 7th International Conference on Learning Representations, ICLR 2019. 2019.
>
> >**[Q2]:** Semanticity is the side effect of the training objective?
>
> The essential idea of "emergence" is that some properties naturally come into being without being explicitly encouraged to do so, hereby the "side effect" as you mentioned. Semanticity is the property we want to evaluate when new graphical conventions emerge, and we did not design designated terms within the model, the game objective, or the training surrogate to direct agents' evolution. Rather:
> (i) The modeling of the game (Sec 3) and the agents (Sec 4.1, 4.2) depicts the probabilistic graphical model and the functional form of the distributions.
> (ii) The objective of the game (Eq(4)) only concerns whether the communication is successful or not and prefers a shorter conversation length.
> (iii) The training surrogate (Eq(5)) is generically derived to (1) better balance the bias and variance according to [39] and (2) make the policy gradient easy to get. These are independent from the modeling and serve only for stable and efficient training.
>
> Note in Fig. 5 that our observation is not that sketches evolve to become "semantic"; the unevolved sketches are already somewhat "semantic," and the evolution **preserves** such semanticity without explicit guidance. Therefore, such a preservation of semanticity validated our model, our objective, and our training method.
>
> >**[Q3]:** A summarized sentence on why the observation spaces between sender and receiver are disjoint.
>
> Thanks for your suggestion. We added it in the revision(L123).
>
> >**[Q4]:** Figure 1 illustration.
>
> Fig 1 shows the process of how the sketches evolve from high iconicity to a more abstract drawing that corresponds to the conversation lengths change in Fig 3(b): the agents first use the detailed sketches to ground the concept. Note that the red color indicates the newly drawn stroke. The receiver can't distinguish between the rooster and the bird until this stroke for crown is drawn (Ronud I). Then for communication efficiency, the sender evolves to draw more abstractly without a clear awareness of the similarity between bird and rooster (Round II). This process introduces some ambiguity. The ambiguity is finally resolved with the evolution goes on (Round III).
>
> >**[Q5]:** How "good" the distractor images are?
>
> Thanks for your suggestion. Communication context is an important topic that we can expand more in future work. Here are some preliminary results that can show the influence of the context distribution on the evolved conventions. We add an additional setting where the context images are sampled based on visual feature (extracted from VGG) similarity (vis-sim). Under this setting, images of high visual similarity with the target image will be more likely to be sampled as the context. As the results shown below, vis-sim can retain higher iconicity while maintaining similar level of symbolicity with the presented complete setting.
> | setting     | classification| generalization-trainset |generalization-unseen instance|generalization-unseen class|
> |-------------|-----------------|-----------------|-----------------|-----------------|
> |complete| 79.44 $\pm$ 0.01|98.07 $\pm$ 0.01(1.03) | 70.37 $\pm$ 0.04(2.36)|  39.40 $\pm$ 0.05(3.76)
> |vis-sim |79.45 $\pm$ 0.01|99.34  $\pm$ 0.01(1.05) | 71.84 $\pm$ 0.03(2.34) | 42.50 $\pm$ 0.01(3.74)

---

> > ### Author Response · Authors · 2022-08-02
> > **Thank you for your review[2/2]**
> >
> > >**[Q6]:** Influence of drawing abilities of the sender?
> >
> > The pretraining of the sender indeed bottlenecks the performance of the evolved agents. Therefore, (1) we have a setting called retrieve that served as an unevolved baseline. Contrasting complete and retrieve, we observed our agents do form "graphical conventions" without sacrificing too much iconicity. (2) There are possibilities that the pretrained sender can use the first 5 strokes to highlight the most informative parts of the target image. But a more interesting phenomenon we want to demonstrate in our experiments is that the sender, though pretrained with random stroke order, can evolve to flexibly adjust the order of the strokes for communication efficiency as shown in fig 6, which is not achieved in the one-step setting when the agents cannot see the complex sketches at the early rounds of the games. (3) In future, we will explore the work mentioned by reviewer b2pi and 6BRM in a sequential setting to alleviate the influence of the pretrained sender.
> >
> > >**[Q7]:** Measuring iconicity with generalizability?
> >
> > Here we show how we get the generalization metric **constructively** from the definition of iconicity. To quantitatively measure the visual similarities between the drawing and the image, we need to compute the distance between the sketch and image in the visual embedding space with cosine similarity:
> > $$Distance(V_{emb(sketch)}, V_{emb(image)}) = \frac{<V_{emb(sketch)}, V_{emb(image)}>}{|V_{emb(sketch)}|| V_{emb(image)}|}$$
> >
> > However, since we do not know the prior of the embeddings, this measure is prone to model-specific biases. Consider the following two scenarios: (1) the embeddings of different images can lie densely in a narrow subspace where $Disctance(V_{emb(image_i)}, V_{emb(image_j)})<\delta$ with a small $\delta$ for any $i, j$, and the embedding of a sketch lies in the middle of the class it belongs and a distractor class; (2) the embedding of different images lie coarsely with $Disctance(V_{emb(image_i)}, V_{emb(image_j)})>>\delta$ with $i, j$ from different classes, and the distance of a sketch to its target class is smaller than the distance to any distractor classes but larger than $\delta/2$. The embedding of case (2) should be better than (1). Yet this **relative** comparison is not reflected in the **unnormalized** value of the raw cosine similarity.
> >
> >   |-------------|------------|     $~~~$ $~~~$ $~~~$      |--------------------------------|----------------|
> >
> >  A  $~~~$  0.5     $~~~$   B  $~~~$ 0.5   $~~~$       C  $~~~$ $~~~$ $~~$A'  $~~~$
> >  $~~~$ $~~~$      4  $~~~$ $~~~$ $~~~$ $~~~$
> > B'  $~~~$ $~~~$ 2 $~~~$ $~~~$C'
> >
> >   $~~~$ $~~~$ $~~~$ $~~~$ (1)  $~~~$ $~~~$ $~~~$ $~~~$ $~~~$ $~~~$ $~~~$ $~~~$ $~~~$ $~~~$ $~~~$ $~~~$ $~~~$ $~~~$ $~~~$ $~~~$(2)
> >
> > Towards this end, we need a normalization mechanism that debiases the metric by taking the global geometry of the embedding space into account. Note that the policy of the receiver takes the form of a softmax of the cosine similarity between the embedding of the sketch and a context set with a target image and some randomly sampled distractors, which naturally approximates the normalization we want. Therefore, the communication accuracy can serve as the visual similarity measure. Since this accuracy is encouraged by the training objective in the seen images, the performance in a held-out subsets of instances / classes (i.e., generalization accuracy) is a natural option to pinpoint the **emergent** preservation of iconicity. We included this explanation in the revised version(L288-295).
> >
> > >**[Q8]:** Hyperparameters.
> >
> > It is the result of a grid search.

---

### Official Review · Reviewer_msus · 2022-07-25

**Rating:** 8
**Confidence:** 4
**Soundness:** 4 excellent
**Presentation:** 2 fair
**Contribution:** 4 excellent

**Summary:**

In this paper, the authors study the problem of emergent communication with sketching. In each game, the sender describes an image through predicting strokes sequentially, while the receiver picks the target image which belongs to the same visual category as the image seen by the sender from a pool of images. The sender is a pretrained stroke predictor and receiver decides whether to make a decision based on the distance between the embeddings of the current canvas and last canvas and the images in the pool. They are jointly  trained with reinforcement learning optimizing for the discounted guessing reward. Note that the canvas renderer is differentiable with respect to the strokes. According to the results, making early decisions is especially useful for the seen instances, since the receiver could learn to adapt to more abstract protocols for the instances it got trained on; the pretrained sender is already a good enough sender which often better than the sender trained in the game, while the latter learns to use less steps in the game; the sequential game is also very crucial since it makes the sender aware of the information gain of each timestep. The authors also evaluate the iconicity, symbolicity and semanticity of drawings. It is also observed in the case study that the sketch becomes simplified and the most silent part of the concept is always preserved.


**Questions:**

According to Section 5.2.2, "iconicity [are defined] as the drawings exhibiting high visual resemblance with the corresponding images", the evaluation I was expecting is on the perceived visual similarities between the drawing and the image. However, the authors merely compared accuracy on the unseen classes. As much as I agree that strong iconicity should implies good generalizability, the other direction does always hold. Can the author explain this mismatch a bit more? I think the authors should directly use generalizability as the property's name or provides human ratings for iconicity. Either way will make the paper much scientifically reliable.

**Limitations:**

The authors have not discussed any potential negative societal impacts. Since the paper uses human sketches as pre-training data, the data might contains the biases, which might be learned by the model. However, I think this is a minor point.

**Strengths And Weaknesses:**

Strengths:

1. This paper studies this very interesting problem of emergent communications of drawing. Language is not the only modality humans have at disposal, and drawing is a more natural way to communicating concrete concepts. Learning to form abstractive conventions will hint us about the way communication protocols are evolved.
2. The technical method and evaluation is sound as a first step towards this question.

Weaknesses:

1. The sender is pretrained and already performs well according to the comparison between sender-fixed and complete models (accuracy-wise). I think a more interesting setup would be training the sender from scratch with limited prior knowledge about the visual categories it needs to depict. This would give us more information about how well a sender can learn to draw with the communication goals.
2. The definition of iconicity and the evaluation are misaligned. Please refer to the questions.

---

> ### Author Response · Authors · 2022-08-02
> **Thank you for your review.**
>
> Thank you for all your valuable feedback and for acknowledging our work studying an interesting problem and proposing sound methods and evaluations.
>
> >**[Q1]:** Pretrained sender vs training sender from scratch.
>
> This work focuses on how the sender can abstract its drawing from high **iconicity**, which necessitates the sender to be pretrained on image-based sketch drawing. The sender-fixed setting is a control setting to investigate the contributions of sender’s evolution to a good graphical convention. Though the sender can achieve high accuracy due to its high iconicity, the low symbolicity and semanticity suggest the importance of mutual adaptation between the two agents.
>
> >**[Q2]:** Measuring iconicity with generalizability?
>
> Here we show how we get the generalization metric **constructively** from the definition of iconicity. To quantitatively measure the visual similarities between the drawing and the image, we compute the distance between the sketch and image in the visual embedding space with cosine similarity:
> $$Distance(V_{emb(sketch)}, V_{emb(image)}) = \frac{<V_{emb(sketch)}, V_{emb(image)}>}{|V_{emb(sketch)}|| V_{emb(image)}|}$$
>
> However, since we do not know the prior of the embeddings, this measure is prone to model-specific biases. Consider the following two scenarios: (1) the embeddings of different images can lie densely in a narrow subspace where $Disctance(V_{emb(image_i)}, V_{emb(image_j)})<\delta$ with a small $\delta$ for any $i, j$, and the embedding of a sketch lies in the middle of the class it belongs and a distractor class; (2) the embedding of different images lie coarsely with $Disctance(V_{emb(image_i)}, V_{emb(image_j)})>>\delta$ with $i, j$ from different classes, and the distance of a sketch to its target class is smaller than the distance to any distractor classes but larger than $\delta/2$. The embedding of case (2) should be better than (1). Yet such **relative** comparison is not reflected in the **unnormalized** value of the raw cosine similarity.
>
>   |-------------|------------|     $~~~$ $~~~$ $~~~$      |--------------------------------|----------------|
>
>  A  $~~~$  0.5     $~~~$   B  $~~~$ 0.5   $~~~$       C  $~~~$ $~~~$ $~~$A'  $~~~$
>  $~~~$ $~~~$      4  $~~~$ $~~~$ $~~~$ $~~~$
> B'  $~~~$ $~~~$ 2 $~~~$ $~~~$C'
>
>   $~~~$ $~~~$ $~~~$ $~~~$ (1)  $~~~$ $~~~$ $~~~$ $~~~$ $~~~$ $~~~$ $~~~$ $~~~$ $~~~$ $~~~$ $~~~$ $~~~$ $~~~$ $~~~$ $~~~$ $~~~$(2)
>
> Towards this end, we need a normalization mechanism that debiases the metric by taking the global geometry of the embedding space into account. Note that the policy of the receiver takes the form of a softmax of the cosine similarity between the embedding of the sketch and a context set with a target image and some randomly sampled distractors, which naturally approximates the normalization we want. Therefore, the communication accuracy can serve as the visual similarity measure. Since this accuracy is encouraged by the training objective in the seen images, the performance in a held-out subsets of instances / classes (i.e., generalization accuracy) is a natural option to pinpoint the **emergent** preservation of iconicity. We included this explanation in the revised version(L288-295).

---

### Meta-Review · Area_Chair_D3qd · 2022-08-24

**Recommendation:** Accept
**Confidence:** Certain

**Metareview:**

This is a solid work, and all reviewers agree to accept the paper.

**Award:**

No

---

### Decision · Program_Chairs · 2022-09-14

Accept